# Isolation of Cell-Free miRNA from Biological Fluids: Influencing Factors and Methods

**DOI:** 10.3390/diagnostics11050865

**Published:** 2021-05-11

**Authors:** Olga Bryzgunova, Maria Konoshenko, Ivan Zaporozhchenko, Alexey Yakovlev, Pavel Laktionov

**Affiliations:** 1Institute of Chemical Biology and Fundamental Medicine, Siberian Branch, Russian Academy of Sciences, 630090 Novosibirsk, Russia; msol@ngs.ru (M.K.); 05alex98@mail.ru (A.Y.); lakt@niboch.nsc.ru (P.L.); 2Meshalkin Siberian Federal Biomedical Research Center, Ministry of Public Health of the Russian Federation, 630055 Novosibirsk, Russia; 3Department of Molecular Biology and Genetics, Aarhus University, 8000 Aarhus, Denmark; ivanzap@niboch.nsc.ru

**Keywords:** extracellular vesicles, lipoproteins, binding to blood cells, cell-free miRNA structure, miRNA storage, isolation kit, miRNA isolation

## Abstract

A vast wealth of recent research has seen attempts of using microRNA (miRNA) found in biological fluids in clinical research and medicine. One of the reasons behind this trend is the apparent their high stability of cell-free miRNA conferred by small size and packaging in supramolecular complexes. However, researchers in both basic and clinical settings often face the problem of selecting adequate methods to extract appropriate quality miRNA preparations for use in specific downstream analysis pipelines. This review outlines the variety of different methods of miRNA isolation from biofluids and examines the key determinants of their efficiency, including, but not limited to, the structural properties of miRNA and factors defining their stability in the extracellular environment.

## 1. Introduction

The biology of miRNA and their transport from cells and into intracellular space, including biological fluids, were actively studied in the last two decades. High stability and value of cell-free miRNA (cf-miRNA) suggests them as promising diagnostic and prognostic biomarkers for a plethora of diseases, including cancers of different origin [1,2,3].

Accurate evaluation of cf-miRNA profiles includes purification, quantification and intelligent data analysis (Figure 1).

The main features and efficacy of isolation techniques are hugely determined by the biological properties of miRNAs, their interaction with other biomolecules and packaging. In turn, purity and quality of retrieved cf-miRNA affect accuracy, reproducibility and reliability of their quantification.

miRNAs are small non-coding RNA molecules (19–22 nt) which can be vastly different in GC-content, base and backbone modifications (Table 1).

Small amount of miRNA is detected in the extracellular space and in biological fluids (see Figure 1 in [4]). Cf-miRNA stability in blood and other biofluids can be attributed to the protection from RNAses conferred by interaction with biomolecules as well as packaging in membrane-coated or membrane-free particles (Table 2).

**Table 1 diagnostics-11-00865-t001:** Features of the structure of cell-free miRNAs (possible variants of chemical modification).

Chemical Modifications	Reference
3′-uridylation	[5,6,7]
adenylation	[5,7]
N6-methyladenosine (m6A) modification	[8,9,10]
3′-terminal 2′-O-methylation	[5,8,11,12]
5-methylcytosine (m5C) modification	[8,13]
adenosine-to-inosine editing	[5,6,7,9,14]
pseudouridine (Ψ) modification	[8]

These complexes strongly interfere with the isolation efficacy requiring liberation of miRNA from such structures to ensure effective cf-miRNA isolation and prevent co-isolation of polymerase reaction inhibitors.

The key to successful study of cf-miRNAs by high throughput methods or precision techniques in research or clinical environment is the quantitative isolation of miRNAs from samples of biological fluids independent from their primary structure, modifications and content of biological fluids. Meanwhile, methodological aspects defining the efficacy and reproducibility of cf-miRNA purification and sample preparation often attract little attention, but in many real-life cases, diagnostically relevant margin of difference is rather narrow and, thus, accurate and reproducible pre-analytical approaches can significantly improve comparative analysis of miRNA expression. In this review, we provide an analysis of current methods of cf-miRNA purification from biological fluids and examine factors that affect its efficacy, including intrinsic features of miRNAs, packaging and common contaminants of miRNA preparations.

## 2. Overview of cf-miRNA Properties and Factors Influencing Their Extraction from Biofluids

Depending on the cell type it can contain up to 120,000 total mature miRNA molecules [52]. This molecular population is heterogeneous, with each miRNA containing (sometimes multiple) miRNA isoforms (isomiRs), different in the sequence of 5′- or 3′-ends [53,54,55]. Additionally, immature miRNA species that may be present in the sample and should be excluded from quantification or analysis. Like other RNA species miRNA can carry a repertoire of base and backbone modifications, including methylation, uridinylation, adenylation, adenosine-to-inosine editing by RNA-dependent adenosine deaminase (ADAR) or inclusion of pseudouridines (Table 1) [5,6,7,11,56,57,58,59,60]. These modifications can additionally affect the half-life of specific miRNAs in biofluids by giving them increased resistance to exonucleases and higher affinity to miRNA binding biomolecules. For example, methylated miR-21-5p is more resistant to digestion by 3′→5′ exoribonuclease polyribonucleotide nucleotidyltransferase 1 (PNPT1) and has higher affinity to Argonaute-2 (AGO2), which may contribute to its higher stability and stronger inhibition of programmed cell death protein 4 (PDCD4) translation, respectively [12]. Since cf-miRNAs originate from the cellular pool one can suppose that they also share the same structural features as cellular miRNA.

The lifetime of free RNAs in biological fluids is very short (15 s in blood according to [61]). This is due to the intrinsic lability of RNA structure, especially in the presence of bivalent metal cations, [62,63] and due to high levels of ribonuclease and phosphodiesterase activity found in most biofluids [64,65,66,67,68,69]. Sequences with lower GC content and stable secondary duplex structures appear to be less stable and therefore at risk of being lost during the extraction process [70]. Despite this, endogenous RNA was shown to survive in bloodstream significantly longer—from several minutes to several hours or even days [71]. Cell-free miRNAs are relatively stable in blood, urine, cerebrospinal liquid, bronchoalveolar lavage, lymph, saliva, milk, tears and others resisting degradation at room temperature for up to 4 days and could withstand as boiling, multiple freeze-thaw cycles, high or low pH [72,73,74]. The observed stability is provided by complexing with proteins, lipoproteins, supramolecular complexes and packaging in membrane-coated vesicles and membrane-free particles (Table 2).

Particle-free miRNAs are frequently found in complexes with proteins: such miRNAs like miR-16 and miR-92a, are mainly (up to 95%) co-precipitated with the protein fraction [44]. In the cell, the main partners of miRNA are the Argonaute proteins (AGO1–4 in humans). It is therefore not surprising that cf-miRNA in supernatants of MCF-7 cells, blood [43], urine [75] and pericardial fluid [76] were found in strong complexes with AGO2 (Kd ~20–80 nM) [77,78,79]. qRT-PCR profiling of 375 miRNAs in size-exclusion chromatography fractions of human plasma more than 67% of the assayed miRNAs were associated with fractions containing AGO2 protein and a corresponding portion of plasma miRNAs could be recovered by AGO2 immunoprecipitation from plasma [47]. In addition to AGO2, miRNA in blood can be in a complex with AGO1 [45].

In blood cf-miRNAs were also found in small complexes 30–40 kDa containing nucleophosmin (NPM1) [46] passing dialisis membranes, although this interaction found in vitro is yet to be confirmed to exist in vivo [43,49]. In other biofluids other interaction partners could be more prevalent, for example in urine Tamm–Horsfall protein (THP; uromodulin) was shown to possibly bind miRNAs [80]. Since number of miRNAs in cell dramatically exceed number of AGO proteins (14 times) significant part of mature miRNAs can be bound with c RISC [81,82,83] and other proteins with classic RNA-binding motifs like HuR, AUF1, etc. [79], involved in transport and functioning of miRNA [84,85].

High density and, to a lesser extent, low density lipoproteins (HDL and LDL, respectively) are another type of confirmed miRNAs transporters in blood, carrying distinct populations of miRNA [38,39,40]. HDL and LDL are 5 to 1000 nm supramolecular complexes (nano- or microparticles) composed of lipoproteins and an assorted collection of lipids [86,87]. Despite the high concentrations of HDL and LDL, it is estimated that they contain no more than 10% of cf-miRNA detectable in blood plasma [39].

Substantial part of cf-miRNA in blood, urine and other biological fluids are packed in membrane-coated extracellular vesicles (EV) which are secreted by normal and cancer cells [88,89,90,91,92]. EV is a collective term describing 30–1000 nm particles coated by a double membrane layer, including exosomes, microvesicles and apoptotic bodies which differ in size, structure, surface markers, molecular composition including distinct miRNAs subsets (Table 2) [15,20,21,93]. On average, one milliliter of blood and urine contains 10^8^–10^12^ or 3–8 × 10^9^ exosomes, correspondingly [94,95]. Some reports suggest that exosomes can be the main miRNA transporters in blood and saliva [31]. Exosomes derived from 4 mL of blood serum typically yield approximately 2–10 ng of RNA; exosomes derived from 10 mL urine yield approximately 2–4 ng RNA, while whole blood serum and urine contain about 10-fold more RNA [94]. Thus, a substantial fraction of cf-miRNA must be associated with proteins, complexes and vesicles other than exosomes. Stoichiometric analysis has shown that the miRNA content of exosomes is not as high as previously thought, with no more than a copy of any single miRNA per exosome, in average [96]. It should be noted that most of miR-16 and miR-223, packed in exosomes, were found to be in complexes with AGO2, adding more fuel to the debate of the dominant form of miRNA circulation in blood [97].

Molecular pattern of apoptotic bodies is not so characteristic like exosomes [16] and can include a variety of proteins, fragments of genomic DNA, most types of RNA including mature and immature miRNA [15,98]. Thus, analyzing miRNA isolated from apoptotic bodies special care should be put into managing potential background. Frequently EV isolation is the first step in the cf-miRNA investigation studies. The methods used for EV enrichment are worth a separate for discussion and have been well summarized in a number of reviews [99,100,101].

As it was mentioned before EV contains membrane and thus a set of lipids like phosphatidylserine, cholesterol, ceramide and sphinogolipids in exosomes [102,103] and lysphosphatidylcholines, sphingmyelin and acylcarnitines in microvesicules [93,104,105]. Presence of lipids in EV suggests necessity of their elimination during miRNA isolation process as well as previously mentioned disintegration of miRNA complexes with biomolecules.

Additionally, recent data suggests that a subpopulation of cf-miRNA circulates bound to the surface blood cells (Table 2). This miRNA fraction is perspective as a source of diagnostic markers because the expression of some cell surface bonded miRNAs changes with the development of cancer [27,32].

To date, there is no universally accepted hypothesis on generation of free cf-miRNA pool. Some types of miRNA–bearing complexes could be actively secreted (exosomes) [106], while for others passive leakage from damaged or dying cells is more feasible [107,108]. Another unknown is how, presumably, different clearance rates of different complexes could impact the half-life of different populations of cf-miRNA.

## 3. Handling and Storage of Biological Fluids before miRNA Isolation

A key step in cf-miRNA isolation includes separation of the liquid portion of biofluids from cells and cell debris. This is necessary to prevent the contamination of sample by cell miRNAs. More than half (almost 58%) of diagnostically relevant miRNAs are highly expressed in at least one type of blood cell [109]. Hemolysis results in significant increase in levels of many cf-miRNAs, including, most prominently, the eritrocyte-specific miR-451a [109,110,111,112]. Significant number of epithelial cells is present in urine, saliva, cerebrospinal and other biofluids [113,114,115] and should be removed before they have the chance to contribute to the cf-miRNA by active secretion or passive leakage. That is commonly achieved by sequential centrifugation at low and high speed [19,116,117] with subsequent separation of cell and debris-free supernatants.

Storage conditions of biofluids samples before removal of cell debris and subsequent storage of blood plasma/serum, cell-free urine and other fluids affect the efficacy of cf-miRNA isolation and quantification [118,119]. Use samples preserved in the slurry resin [119], long-term storage may decrease miRNA concentration in the sample as do multiple freeze-thaw cycles [4,118]. As described in the review [4] for midterm storage (<20 month) no major differences in serum miRNA levels were observed between −80 °C and −20 °C, nonetheless some individual miRNA were seriously affected by those conditions. However, a slightly decrease within the range of 2–4 years; after 6 years of storage, a significant decrease of miRNA levels was perceived that only accentuates in the course of time.

Concerning dried serum spots incomplete drying of blots before storing was prejudicial for its preservation [4].

In experiments with urine all conditions demonstrated a surprising degree of stability of miRNAs: by the end of ten freeze–thaw cycles, 23–37% of the initial amount remained; over the 5-day period of storage at room temperature, 35% of the initial amount remained; and at 4 °C, 42–56% of the initial amount remained [120].

With that, it is possible that different fractions of miRNA, including particle-free miRNA and EV cargo miRNA can differ in sensitivity to storage conditions. For example, exosomal miRNAs showed extra stability under different storage conditions [121]. Stability of individual miRNA in general also depends on the storage conditions. For example, miR-145 and miR-20a degraded at room temperature, but both are stable at 4 °C, −20 °C and −80 °C for 72 h in serum and as cDNA, which was additionally shown to be stable for at least 3 months at −20 °C and survive four freeze-thaw cycles at −20 °C without significant degradation [122].

Based on the foregoing, it may be concluded that not all biofluids are similar in terms of storage conditions and the method and time of storage should be determined by the tasks set in each specific case.

## 4. miRNA Isolation from Biofluids

### 4.1. General Considerations and Technical Parameters Defining Isolation Method

While choosing a method for isolation of miRNAs from biological fluids, it is necessary to take into account the peculiarities of their composition (ionics, proteins, polysaccharides, etc.), which can affect the efficiency of this isolation. For example, blood plasma contains high amount of proteins with total concentration of 7.2 g/dL forhealthyadult human [123]. Accordingly, the concentration of chaotropic agents should be sufficient to extract miRNAs from various protein complexes. Urine contains a large fraction of nitrogen, with urea the most predominant, phosphorus, sodium and potassium, with the total suspended solids at 21 mg/L and total dissolved solids at 31.4 mg/g [124]. That is why, while using chaotropic salts (guanidine), their concentration should be lower than for isolation of miRNA from blood plasma. Normal saliva contains a large amount of glucose (0.5–1.00 mg/100 mL) [125], which can also affect the efficiency of excretion.

Progress in cf-miRNA extraction technologies has made it evident that any effective protocol should successfully achieve three main goals:− enable complete dissociation of miRNA complexes with biomolecules of different nature;− protect miRNA from enzymatic and non-enzymatic degradation during isolation regardless of their sequence;− prevent contamination of miRNA preparations with inhibitors of enzymes used in downstream analyses or substances that hinder accurate quantification by UV absorption or fluorescence detection.

While numerous protocols allow for total RNA isolation, several more recent options are specifically tailored for miRNA isolation and provide either isolation of total RNA without loss of miRNA or selective purification of miRNA [70]. The choice of isolation method not only determines the quality and quantity of extracted miRNA as a whole, but can also favor isolation of certain individual cf-miRNAs leading to differences in the expression of the same miRNAs when isolated by different methods. Extraction of at least some cf-miRNAs is closely dependent on the extraction method suggesting incomplete dissociation of cf-miRNA complexes and highlighting a connection between the efficacy of isolation, miRNA sequence and type of complexing with biomolecules [126].

For example, depending on the starting volume of the biological samples used, the conditions of the protocol can favor the extraction of GC-poor miRNAs [127]. Such differences in isolation efficiency can have a dramatic impact on downstream analyses and most importantly normalization of cf-miRNAs.

Successful liberation of miRNA from complexes is the key to efficient miRNA isolation. Formation of complexes with proteins and lipoproteins can be sequence dependent. Complexes of different type can have different affinities and stabilized by different types of interactions (ionic, hydrophobic, etc.). EV contain membrane featuring distinct combinations of lipids: phosphatidylserine, cholesterol, ceramide and sphinogolipids in exosomes [102,103]; lysphosphatidylcholines, sphingmyelin and acylcarnitines in microvesicles [93,104,105]. High lipid content of EV and lipoproteins needs to be removed during miRNA isolation and any possible interaction with miRNA should be disrupted.

Complete dissociation of heterogeneous complexes in the presence of high abundance of biomolecules of varied nature often demands using high excess of denaturing solutions. This is true for blood plasma, serum and especially urine. To ensure adequate denaturation and removal of the high protein content from samples (albumin, immunoglobulins, coagulation and complement components among others), the lysis reagent-to-specimen ratio has to be increased several-fold. Together with the starting fluid volume, this is the most variable step in different protocols [128] leads to use large volume of pelleting reagents or increasing of efficacy of miRNA binding by adsorbents.

When planning a miRNA study, it is necessary to choose an extraction method taking into account the following parameters: sample type, expected type of miRNA packaging and its abundance, compatibility of the method with downstream applications (for example, with the method of EVs isolation), the duration and cost of the procedure, the available infrastructure (e.g., ability to safely handle and dispose of hazardous substances.

### 4.2. Methods of miRNA Isolation from Biofliuds

The overwhelming majority of currently used methods for miRNAs isolation from biological fluids are based on acid guanidinium thiocyanate-phenol-chloroform extraction, pioneered by Chomczynski and Sacchi in 1987 [129,130]. This method allows efficient fractionation of RNA, DNA and proteins and dissociates most part of miRNA complexes, which is why it is considered the “gold standard” for total RNA (and by proxy miRNA) extraction. A number of commercial kits, for example, Trizol LS, utilize phenol-based extraction for isolation of miRNAs in a fast and simplified manner and can also further specialize in dealing with specific sample types, such as serum and plasma, animal or plant tissue [70]. Products like miRVANA and miRNeasy include an additional stage of purification on columns with fiberglass sorbents for miRNA enrichment [131]. Solid phase extraction methods take advantage of the interactions between the functional groups of nucleic acids and solid sorbents under particular conditions. The adsorption of nucleic acids on the silica surface can be regulated with the use of chaotropic agents at different pH, temperature or ionic strength and additionally enhanced by the addition of bivalent metal ions into the sorption buffer or the sorbent itself [132]. The efficiency of elution also depends on pH and temperature, although recently RNAse-free water is the elution buffer of choice, given its convenience for most downstream applications.

A number of comparative analyses show that methods based on guanidinium thiocyanate-phenol-chloroform extraction are superior in terms of the efficiency, but different versions of the protocol differ in performance (Table 3) [133]. 

However, with all of the advantages, this method has some significant limitations. The procedure is time-consuming (40–60 min) and poorly scalable to the starting volume of the sample, requires high quality reagents (phenol). In addition, phenol is highly toxic; thus, protective equipment (flow hood) and chemical waste management systems are required to minimize impact on personnel health and environment. The disadvantages of the method also include the partial loss of RNA with a low GC content [149]. All of the above mentioned creates difficulties for its routine use in many scenarios, including clinical setting [150].

To date, several strategies have been suggested as replacement for acid phenol-chloroform extraction. Strong chaotropic properties of guanidinium thiocyanate allow efficient dissociation of both cell-free DNA and RNA complexes. However, some cf-miRNA complexes seem to resist the effects of guanidinium thiocyanate [151]. We previously suggested using Folch solution combined with guanidinium thiocyanate to disrupt hydrophobic interactions and assure complete dissociation of miRNA-containing complexes with subsequent application of the mix directly to fiberglass sorbents [151]. The method avoids bypasses using phenol or detergents, which is undesirable due to micellation and subsequent reduction in the efficiency of fiberglass sorbent. The method was shown to successfully retrieve miRNA from blood plasma, but has proven cumbersome for cf-miRNA extraction from urine due to the large starting volume.

In the mercury LNA RT kit (Qiagen, Hilden, Germany) instead of a laborious organic extraction procedure, cf-miRNA complexes are dissociated and excess protein and lipoprotein content is precipitated while miRNA is purified from the supernatant using fiberglass columns [133]. Extraction only requires 30 min and avoids highly toxic chemical agents. According to several studies, this approach is at least as effective as phenol-chloroform extraction [131,152]. Possible drawbacks of the methods could include partial loss of miRNA as a result of incomplete denaturation of the complexes or co-precipitation with excess biopolymers present in biological fluids. There is also evidence that an additional phosphorylation step may be required to use the obtained preparations for miRNA sequencing analysis [131,152].

Another method based on precipitation of biopolymers with octanoic acid was offered for immunoglobulin isolation [153]. In our lab, we found that complexes of miRNA in blood and urine were dissociated and pelleted by octanoic acid in the presence guanidinium thiocyanate, followed miRNA cleanup with fiberglass columns [154]. The method has demonstrated good efficacy of miRNA isolation from blood and even better performance in isolating urine cf-miRNA. Meanwhile, disadvantages of the method have not been fully explored yet.

Some isolation techniques are to be used when special downstream analyses are planned. For example, the most commonly used method to study protein/RNA complexes is co-immunoprecipitation of RNA with antibodies against the protein interacting with it [155]. Immunoprecipitation can be conducted when antibodies are added to solution and then mixed with the antibody sorbent in the resulting mixture, or an antibody is immobilized to a sorbent and then added to a solution of the protein. These approaches allow to study miRNA-AGO2 complexes [48]. It is known that immunoprecipitation which differ in the order of interaction of the components of the complex, (antigen (Ago2) antibodies and PA-sepharose) as well as in presence of blocking antibodies favor different miRNAs [48]. The authors suggest that non-specific binding to PA-sepharose and autoantibodies against miRNA binding proteins might contribute obtained results [48].

Lipoprotein-bound miRNAs can be isolated using sequential density ultracentrifugation of plasma with the adjustment to well-defined densities using potassium bromide salt. Another method used to separate plasma lipoproteins is the size exclusion chromatography, often with a fast-protein liquid chromatography (FPLC) system combined with columns filled with high-resolution stationary phases [156]. Affinity chromatography using columns linked to monoclonal antibodies specific to apolipoproteins offer one more approach of plasma lipoprotein isolation. However, the use of low pH required to elute lipoproteins retained by the immuno-affinity column lead to potential risk for the loss of nucleic acids from the purified lipoprotein. Thus, sequential density ultracentrifugation remains the standard for the isolation of well-defined lipoprotein classes in sufficient quantities to conduct subsequent in vitro or in vivo studies of lipoprotein function and RNA composition [156].

Finally, there are methods and kits designed used for miRNA isolation from sources other than biofluids, which could be adopted for cf-miRNA. For example, RNAgem (microGEM, Southampton, UK) provides temperature-driven, single-tube extraction of total RNA and miRNA from mammalian cells, tissues, insects, bacteria and virus.

The presence or the absence of the precipitation of miRNA and the type of the co-precipitator may affect the miRNA yield during miRNA isolation. The re-precipitation stage represents an additional stage of miRNA sample processing which lead to longer time of isolation and to potential loss of miRNA. The similar problem is known for circulating DNA precipitation, for which in unpredictable DNA loss or contamination was shown for some precipitation protocols, especially those which used positively charged compounds [157]. RNA bacteriophage carrier (MS2), yeast RNA, tRNA and glycogen are usually used carriers [137,158,159,160]. The most common co-precipitator is glycogen; however, the combination of tRNA and glycogen was shown to improve the yield and purity of RNA greatly and to maximize the extraction of miRNA from plasma when using the TRIzol LS [160]. In our opinion, the most suitable carrier is glycogen, since it does not contain components of a nucleic nature which may interfere subsequent analysis of miRNAs (NGS, microchip technology, etc.).

Numerous efforts have been taken to compare the effectiveness of different miRNA isolation methods (Table 3). Most of these studies compare protocols for miRNA isolation from blood plasma and most commonly simple phenol-chloroform extraction is superior to both phenol extraction with column-based cleanup techniques and phenol-free column-based methods (Table 3). No conclusive results have been obtained regarding the differences in the performance of column-based methods, suggesting that an even greater effort is still needed to compare existing extraction methods and working toward developing universal standards for miRNA extraction (Table 3; [116]).

Rapidly developing microfluidic technologies could further enhance cf-miRNA extraction. Microfluidic devices are compact units composed of a network of microchannels with diameters of tens to hundreds of micrometers capable of handling viscous media within a concentration range of pico- to microliters. Specialized units are used for tuning of fluid movement. Microfluidic devices have a tremendous potential and are able to reproduce laboratory techniques on a microscale with a high accuracy and specificity [161]. MiRNA isolation as well as their detection is among areas of application of microfluidic technologies. Recently developed microfluidic platforms can perform effective exosome separation and exosomal miRNA detection for liquid biopsies within a single device. Such methods offer advantages of integrity and fast procedure [162]. Moreover, in microscale processes reagent consumption can be reduced from milliliters to microliters. [161].

## 5. Conclusions

Current data clearly show there is no single most effective method of cf-miRNAs isolation and the choice of methodology is often dictated by sample type and properties of the miRNA fraction under investigation, as well as the ease of use in the context of each specific research project. This strongly aligns with the opinion of NIH Extracellular RNA Communication Consortium, which states that, despite the significant development of the methodology, there is no optimal method for isolating miRNAs and work on the development and optimization of new approaches to this problem must be continued [152]. At this point, both attempts to refine and optimize existing technologies and exploration of novel approaches could shift this paradigm and give a significant boost to cf-miRNA studies and potentially see some of their diagnostic applications make an appearance in the clinical setting.

## Figures and Tables

**Figure 1 diagnostics-11-00865-f001:**
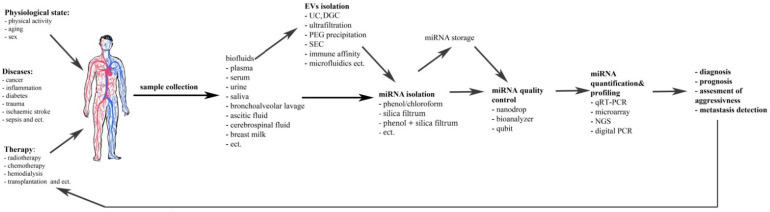
The scheme of cell-free miRNA investigations and their possible implementation.

**Table 2 diagnostics-11-00865-t002:** Forms of cell-free miRNAs.

Forms of Binding	References
Membrane-coated microparticles	
Apoptotic bodies (0.1–5 µm)	[15,16,17]
Microvesicles (>0.5 µm)	[17,18,19,20,21,22,23]
Ectosomes (100–600 nm)	[24]
Oncosomes (1–10 µm)	[17,25,26]
Exosomes (30–150 nm)	[17,18,22,24,27,28,29,30,31,32,33]
Specific microvesicles (prostasomes (50 nm–0.5 μm), melanosomes (>0.5 µm), ‘platelet dust’ (~130–500 nm))	[33,34,35,36,37]
Membrane-free microparticles	
High-density lipoproteins (HDL)	[23,38,39,40,41,42]
Low-density lipoproteins (LDL)	[39,40,42]
Various RNA-binding proteins (for example, AGO1, AGO2, nucleophosmin 1 (NPM1), Tamm-Horsfall protein (THP), etc.)	[23,43,44,45,46,47,48,49]
Exomeres (~35 nm (<50 nm))	[50,51]
miRNAs associated with the surface of blood cells	[27,32]

**Table 3 diagnostics-11-00865-t003:** Comparative studies of different miRNA isolation methods.

The Performance of Different Isolation Methods in Terms of Yield, Purity and miRNAs Detection Level	miRNAs & Detection Method	Sample Type	References
Total RNA isolation kit < miRNeasy kit, the miRVana PARIS kit	miRCURY LNA real-time PCR panel	serum	[134]
TRIzol-based miRNA isolation<The miRNeasy Mini kit	qRT-PCR miR-92a, -126	plasma, serum	[135]
miRvana microRNA Isolation kit, QIAamp Circulating Nucleic Acid Kit > miRNeasy Serum/Plasma Kit	qRT-PCR miR-21, -191, syntheticath-miR-159a	plasma	[136]
miRNeasy Serum/Plasma kit, MagnaZol™ cfRNA Isolation Reagent	NGS	plasma	[137]
miRCURY Cell and Plant Kit without glycogen < TRIzol LS (LT) followed by mirVana column clean-up with glycogen/without glycogen< miRCURY Cell and Plant Kit with glycogen < mirVana kit with glycogen/without glycogen miRCURY Cell and Plant Kit with glycogen < miRCURY Biofluids kit with glycogen/without glycogen miRNeasy Serum/Plasma system slightly < miRCURY Biofluids kit	qRT-PCR miR-16, -21, cel-miR-39	plasma	[131]
TRIzol LS (LT); miRNA purification kit < mirVana PARIS kit	qRT-PCR cel-miR-39	plasma	[138]
TRIzol LS (LT); miRNeasy mini kit; mirVana PARIS kit; miRNA purification kit	qRT-PCR miR-21		
mirVana PARIS, Trizol LS (LT) with mirVana, miRCURY < miRNeasy, < miRNeasy with RNeasy MinElute Cleanup Kit	Bioanalyser	urinary exosomes	[106]
mirVana PARIS (with addition of an additional organic extraction step), < miRNeasy kit	qRT-PCR	plasma serum	[139]
comparable results: miRNeasy mini kit, Plasma/Serum Circulating and Exosomal RNA Purification Kit = NucleoSpin miRNAs Plasma kit	Microarray	peripheral blood mononuclear cells	[127]
MirVana PARIS kit, TRIzol-LS (LT) < miRNeasy Serum/Plasma Kit		fresh plasma	
TRIzol-LS (LT) < mirVana PARIS kit, TRIzol-LS following mirVana kit < miRNeasy Serum/Plasma kit		frozen plasma	
comparable results in terms of Cq values: miRCURY RNA Isolation Kits, Plasma/Serum Circulating and Exosomal RNA Purification Mini Kit, NucleoSpin miRNA Plasma, miRNeasy Serum/Plasma Kit, Direct-zol RNA MiniPrep	miRCURY microRNA QC PCR Panel (Exiqon) miR-18a, -21, -29a	plasma	[140]
TRIzol-LS (LT) < miRNeasy, mirVana miRNA Isolation Kit < TRIzol-LS (LT) combined with miRNeasy < miRCURY RNA Isolation Kit < SeraMir exoRNA columns	RT-qPCR, mir-16 and let-7i	UE purified by UF or from peripheral blood mononuclear cells	[141]
PAXgene and Tempus whole blood RNA preservation tubes Qiagen (PAXgene Blood miRNA Kit), Life Technologies (MagMAX for Stabilized Blood Tubes RNA Isolation Kit), Norgen Biotek (Norgen Preserved Blood RNA Purification Kit I and Kit II) and 2 (semi-) automated protocols on the QIAsymphony (Qiagen) and MagMAX Express-96 Magnetic Particle Processor (Life Technologies) Norgen Kit II with PAXgene < Norgen Kit I with Tempus Tubes	qRT-PCR miR-1, -16, -30b, -133a	blood	[140]
QIAGEN Exo kit < UC fraction using the miRNeasy kits and the NucleoSpin kit	miRCURY microRNA QC PCR Panel (Exiqon) miR-103, -191, -23a, -451, -18a, -21, -29a	EVs	[142]
Similar Cq values: miRNeasy Mini Kit, miRNeasy Serum/Plasma kit, exoRNeasy Serum/Plasma Starter Kit, Plasma/Serum RNA Purification Mini Kit, Direct-zol™ RNA MiniPrep, NucleoSpin miRNA Plasma		plasma	
miRNeasy slightly better than miRCURY	RT-qPCR, mir-16, -106a, -222, -223	plasma exosomes	[133]
miRCUR < miRNeasy miRCURY biofluids = miRNeasy kits	RT-qPCR, mir-30c-2-3p, -106a, -204, -222, -141	plasma	
miRCURY < miRNeasy		urinary exosome	
miRNeasy Serum/Plasma kit, Plasma/Serum Circulating RNA Purification Kit < miRCURY RNA Isolation Kit—Biofluids; mirVana PARIS Kit; NucleoSpin miRNA Plasma	TaqMan miRNA PCR system (Applied Biosystems, California, USA), miRCURY LNA microRNA PCR system (Exiqon) and miScript miRNA PCR system (Qiagen), miR-16, -141, -135a	plasma	
PureLink commercial column extraction kit < TiO2 nanofibers	qRT-PCR miR-21, -191, spiked cel-miR-54	MDA-MB-231 breast cancer cells	[133]
Phenol and column-based procedure and a column-based procedure, in the presence or absence of two RNA carriers (yeast RNA and MS2 RNA); others carriers and their absence < yeast RNA	NGS	plasma	[143]
ExoRNeasy Serum/Plasma Starter Kit (Cat. 77023), Plasma/Serum RNA Purification Mini Kit, Direct-zol™ RNA MiniPrep < miRNeasy kits (Mini and Serum/Plasma kits) and NucleoSpin kit	Bioanalyzer, qRT-PCR miR-122, -21-1, -30d, -451a	plasma	[144]
Trizol LS reagent isolation method < miRNeasy mini kit by 35%	qRT-PCR RNU6, miR-145, -20a	serum	[145]
Trizol LS < miRCURY < miRNeasy kit, with some peculiarities concerning different sample volumes	NGS, qRT-PCR miR-106a, -222, -16, -223	plasma and circulating exosomes	[146]
mirVana and the Plasma/Serum RNA Purification Mini < miRNeasy Serum/Plasma < miRCURY Biofluids and the miRNeasy Advanced Serum/Plasma	NGS, qRT-PCR let-7a-5p, miR-150-5p, -16-5p, -122–5p, -21-5p, -191-5p, cel-miR-39-3p	plasma cf-miRNA	[122]
exoRNeasy Serum/Plasma Exosome slightly <Ultracentrifugation + miRNeasy kit		plasma EVmiRNA	
Circulating Nucleic Acid Kit < TRIzol LS (TF); miRNEasy; RNA extraction kit and the MiRCURY RNA Isolation Kit—intermediate	NGS	plasma	[147]
miRNeasy Serum/Plasma kit (Qiagen, Germany), miRVANA miRNA Isolation Kit (Ambion, Austin, TX, USA) and TRIzol LS (TF) < Urine microRNA purification kit	qRT-qPCR miR-16, -10a-5p, -196a-5p) and exogenous cel-miR-39	CSF samples	[148]
Quick-cfRNA Serum and plasma; Isolate II miRNA kit; PureLink RNA mini kit; Monarch total RNA miniprep kit < miRNeasy Serum/Plasma Advanced kit < miRNeasy Serum/Plasma kit	qRT-PCR miR-19b, -92a, -93, -103, -144, -345-3p, -486, -494-3p, -1306	ovine plasma, fresh and frozen	[70]

<, >—better or worth in terms of yield, purity and miRNAs detection level. Total RNA isolation kit, (Norgen, Thorold, Ontario, Canada);Plasma/Serum Circulating and Exosomal RNA Purification Mini Kit, (Norgen, Thorold, Ontario, Canada);Urine microRNA purification kit, (Norgen, Thorold, Ontario, Canada);MiRNeasy kit, (Qiagen, Hilden, Germany);QIAamp^®^ Circulating Nucleic Acid Kit, (Qiagen, Hilden, Germany);ExoRNeasy Serum/Plasma Starter Kit, (Qiagen, Hilden, Germany);MiRNeasy Serum/Plasma Kit, (Qiagen, Hilden, Germany);MiRNeasy with RNeasy MinElute Cleanup Kit, (Qiagen, Hilden, Germany);QiaSymphony RNA extraction kit, (Qiagen, Hilden, Germany);MiRVana PARIS (Ambion, Life Technologies, Austin, TX, USA); MiRvana microRNA Isolation kit (Ambion, Life Technologies, Austin, TX, USA ); Trizol LS reagent (LT) (Ambion, Life Technologies, Carlsbad, CA, USA); MiRCURY RNA isolation kit Biofluids (Exiqon, Vedbaek, Denmark); MiRCURY Cell and Plant Kit (Exiqon, Vedbaek, Denmark); MiRCURY Biofluids kit (Exiqon, Vedbaek, Denmark); MagnaZol™ cfRNA Isolation Reagent (Bioo Scientific, Austin, TX, USA ); ThermoFisher Scientific Ambion TRIzol LS Reagent (TF) (Thermo Fisher Scientific, Third Avenue Waltham, MA, USA); PureLink commercial column extraction kit (Thermo Fisher Scientific, Third Avenue Waltham, MA, USA ); NucleoSpin miRNA Plasma kit (Macherey–Nagel, Düren, Germany); Direct-zol RNA MiniPrep (Zymo Research, Irvine, CA, USA); Quick-cfRNA Serum and plasma (Zymo Research, Irvine, CA, USA); SeraMirTM exoRNA columns (System Biosciences, Palo Alto, CA, USA); PureLink RNA mini kit (Bioline, Memphis, TN, USA); Monarch total RNA miniprep kit (NEB, Hitchin, UK).

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
