# Peer review of "Isolation of Cell-Free miRNA from Biological Fluids: Influencing Factors and Methods"

_diagnostics, 2021, doi:10.3390/diagnostics11050865_

Round 1
Reviewer 1 Report
In this manuscript, Bryzgunova et al. provide an overview of factors and methods affecting the isolation of miRNAs from biological fluids. Authors essentially compile an inventory of the literature. Overall, the manuscript is well-structured and carefully prepared.
- In its current form, however, a few things are only mentioned in Table 2 (e.g., ectosomes, oncosomes, exomeres etc). It also needs to describe them in the main text in more detail (e.g., what are they, what miRNAs are present in them etc) for some readers.
- Table 2 does not include everything mentioned in the text (e.g., THP is missing in the table). Instead of using “etc” term, it is better to precisely organize Table contents.
- If using the greater-than sign (>), the less-than sign, and an equal sign in Table 3, the meaning of them is also required to mention at the bottom of table although it can be inferred.
- Typos are here and there, so take this opportunity to check the whole manuscript again.
- Line 120: use superscript for the number of exosomes
Author Response
Dear Reviewer, we were pleased to receive your valuable advice and suggestions on how to improve our manuscript. Your comments were really useful and particularly relevant for the improvement of our article. Please find the answers to your comments below.
Point 1. In its current form, however, a few things are only mentioned in Table 2 (e.g., ectosomes, oncosomes, exomeres etc). It also needs to describe them in the main text in more detail (e.g., what are they, what miRNAs are present in them etc) for some readers.
Thank you for the comment, indeed, there is a great number of different types of extracellular vesicles, moreover, there are different and sometimes contradictory classifications. They were described and discussed in our previous review (Konoshenko et al., 2019 [99]). In present review we would like to limit this information section order to make the review more targeted and clear. Concerning miRNA composition of different vesicles types, up to now there are no common accepted protocols for separation of these vesicles types from each other and thus no definite and unambiguous information on the exact nucleic acid composition of each type of vesicle. However, we have added valuable references (P.1) to make this information more accessible.
Point 2. Table 2 does not include everything mentioned in the text (e.g., THP is missing in the table). Instead of using “etc” term, it is better to precisely organize Table contents.
Thank you for the comment, the THP data was added to Table 2.
Point 3. If using the greater-than sign (>), the less-than sign, and an equal sign in Table 3, the meaning of them is also required to mention at the bottom of table although it can be inferred.
Thank you for the comment, the relevant information has been added under the Table 3.
Point 4. Typos are here and there, so take this opportunity to check the whole manuscript again.
We did our best to correct all the typos, thank you!
Point 5. Line 120: use superscript for the number of exosomes
Yes, we have introduce necessary changes, thank you.
We are grateful to the reviewer for the comments, the corresponding changes were made to the manuscript.
Reviewer 2 Report
In this review article, the authors summarized recent methods of miRNA isolation from biofluids and examined the key determinants of their efficiency. This review brings together a wide body of information that may be of use for researchers interested in this field. My specific comment for this manuscript is listed below.
- Could authors summarize the advantages, disadvantages, and limitations of each method of miRNA isolation from biofluids? It will help readers to choose the proper method easily.
- On page 4, line 120: “108-1012 or 3-8x109” should be “108-1012 or 3-8x109”.
- On page 4, line 128: “that most of miR-16 и miR-223,” should be “that most of miR-16 and miR-223,”.
Author Response
Dear Reviewer, we appreciate your evaluation of our manuscript and thank you for your comments and advices. They were really useful and particularly relevant to the improvement of our article.
Point 1. Could authors summarize the advantages, disadvantages, and limitations of each method of miRNA isolation from biofluids? It will help readers to choose the proper method easily.
We support the desire of the reviewer, but this is quite difficult to do, since different methods may be more effective for different biological fluid. For example, large urine volume demands methods minimally increasing sample volume. Table 3 demonstrates the initial data for selection of the most convenient method from commercial ones. To understand and evaluate better the advantages, disadvantages and limitations of different protocols we introduce first chapters describing factors interfering with isolation efficacy. We prefer not to evaluate/compare methods not to be impolite making a personal estimates especially when there are no direct comparison data (data not entering table 3).
Point 2. On page 4, line 120: “108-1012 or 3-8x109” should be “108-1012 or 3-8x109”.
The corresponding changes were made, thank you!
Point 3. On page 4, line 128: “that most of miR-16 и miR-223,” should be “that most of miR-16 and miR-223,”.
The corresponding changes were made, thank you!
Reviewer 3 Report
The manuscript by Bryzgunova and coworkers describes in a Review format, the current available methods the isolation of miRNAs from biofluids. The topic is relevant and interesting for the journal readers, however, in my humble opinion the manuscript requires a deep re-organization to be clearer and more compelling. Moreover, the distribution of the text, and the design of tables and figures needs a careful revision. Consequently, in my opinion the manuscript is not suitable for publication in this current form, and I would recommend a major revision.
One important aspect is the denomination that the authors selected for the circulating miRNAs, called as cell-free miRNAs or cf-miRNAs. I would advice to use extracellular miRNAs or ex-miRNAs instead, since the “cell free” term is more applicable to total cellular extracts where the insoluble components were removed.
Specific point-to-point topics to be considered:
1.- Figure 1 needs to be improved, increasing the size of the fonts to be clearly readable. Please check also for potential copyright issues related with the reproduction of elsewhere graphical contents.
2.- The location of Tables 1 needs to be modified, probably going down to the first page. Regarding table 2, I sincerely think that the authors may consider deleting it since the contents showed could be included within the text.
3.- Line 42; the authors claim that the miRNAs could interact with other “biopolymers”. I would suggest to replace the word “biopolymer” by “biomolecule”.
4.- I do not agree with the statement mentioned in lines 148-152, where the authors described that there is no consensual idea about the origin of cf-miRNAs (ex-miRNAs). The population of ex-miRNAs generated from cellular leakage is minimum when compared to the miRNAs that are actively secreted by cells. There are excellent evidences in the literature that point to this direction. Moreover, the text also pointed out that the stability of miRNAs is mainly related with their binding to different biomolecules or their encapsulation in membrane-based vesicles. The authors would require to cite and discuss, that miRNAs are very poor substrates for nucleases due to their small size.
5.- Section 3 (handling and storage) appears to be designed for the description of blood-related miRNAs. This section needs to be expanded, considering the ex-miRNAs present in other biofluids to answer the questions: are all the fluids equivalent in terms of storage conditions?, is there any general protocol for the fluid storage prior to miRNA analysis?.
6.- Table 3 needs to be re-designed. In this current form it is very difficult to read and more important, it is almost impossible to realize what the authors want to show. I understand that the subjacent idea is to compile the comparative studies of miRNA extraction to derive some functional information for the reader. In order to achieve this goal, the table could be converted to a graphical scheme where the different methods are represented by symbols with different sizes according to their applicability in different scenarios. Remember that a typical reader of this kind of paper would always look for comparative analysis represented in a very clear and appealing form, easy to read, and to understand. The information should include all the relevant details, avoiding general descriptions of methods such as “total RNA extraction”.
7.- Section 4, should be divided into sub-sections, namely those describing the methods, technical details and applications. In this current text version, the information is very disperse and redundant sometimes.
8.- The authors would need to derive clear conclusions from their study, showing a “golden standard protocol” for extraction of miRNAs from biofluids. It would be advisable to comment also the intrinsic characteristics of some biofluids like saliva, which high glycoprotein content could interfere with the extraction procedure. The manuscript should comment these differences, proposing technical attachments to the general protocol that will be applied to specific biofluids. For instance, it would be advisable to previously remove sugars from the saliva before RNA extraction.
9.- I would also suggest to include a specific subsection related with the so called "carriers". The authors briefly refer the use of glycogen for RNA precipitation, but there are other carriers also frequently employed like yeast tRNA or the MS2 RNA (which are just cited in a specific method described in Table 3, but not discussed alongside the text). What are the advantages of these carriers?. Would the authors advice a naive scientist in the field to use them? In which circumstances?
Author Response
Dear Reviewer, thank you for valuable advices and suggestions on how to improve our manuscript. Your comments were really useful and relevant for the improvement of our article. Please find the answers to your comments below.
Point 1. One important aspect is the denomination that the authors selected for the circulating miRNAs, called as cell-free miRNAs or cf-miRNAs. I would advice to use extracellular miRNAs or ex-miRNAs instead, since the “cell free” term is more applicable to total cellular extracts where the insoluble components were removed.
Indeed, some researches use the term ex-miRNAs for the circulating miRNAs, however, these is a number of studies, which use the same term for exosomal microRNAs (for example, https://pubmed.ncbi.nlm.nih.gov/33093831/ ; https://pubmed.ncbi.nlm.nih.gov/32724386/ ; https://pubmed.ncbi.nlm.nih.gov/31611956/). That’s why we use the term cf-miRNAs to avoid discrepancies. Actually, the term “cell-free nucleic acids” is a common term in the field of liquid biopsy including blood, urine and other biological fluids and this was the second reason to use this abbreviation.
Point 2. Figure 1 needs to be improved, increasing the size of the fonts to be clearly readable. Please check also for potential copyright issues related with the reproduction of elsewhere graphical contents.
We thank the reviewer for the comment, we have increased the size of the figure to make it easier to read. The graphic component of the figure was provided to the paper by its author (Faina Solov'eva), the gratitude to who is expressed by us in the Acknowledgments section so the copyrights have been respected.
Point 3. The location of Tables 1 needs to be modified, probably going down to the first page. Regarding table 2, I sincerely think that the authors may consider deleting it since the contents showed could be included within the text.
Thanks for the suggestion, however, there are no space for table 1 at the first page (final tables positioning is usually done by publisher). We prefer to stay Table 2 because we consider this information more acceptable being introduced not in text but in the table.
Point 4. Line 42; the authors claim that the miRNAs could interact with other “biopolymers”. I would suggest to replace the word “biopolymer” by “biomolecule”.
Thank you, yes, biomolecules is more correct and “biopolymers” was replaced by “biomolecules” in the text.
Point 5. I do not agree with the statement mentioned in lines 148-152, where the authors described that there is no consensual idea about the origin of cf-miRNAs (ex-miRNAs). The population of ex-miRNAs generated from cellular leakage is minimum when compared to the miRNAs that are actively secreted by cells. There are excellent evidences in the literature that point to this direction.
Yes, we have added links that indicate the possible different sources of origin of extracellular miRNAs (P4).
Moreover, the text also pointed out that the stability of miRNAs is mainly related with their binding to different biomolecules or their encapsulation in membrane-based vesicles. The authors would require to cite and discuss, that miRNAs are very poor substrates for nucleases due to their small size.
It is known that RNA stability depend on sequence for different RNAses, but as far as we know there is no data on RNA stability in depend from their size. Stability of ribooligonucleotides is very poor in blood (and other biological fluids, when they do not contain protective groups) [https://pubmed.ncbi.nlm.nih.gov/21028964/ https://faseb.onlinelibrary.wiley.com/doi/full/10.1096/fj.09.142398] and RNA oligonucleotides are quickly digested in vitro and in vivo by endogenous nucleases many manuscripts claim that miRNAs stability is mainly concerned with their interaction with biomolecules. From our own experience (with P32 labelled riboODNs) we know that ribooloigonucleotides (20-25 mers) are not stable against RNAses and chemical hydrolysis, they are partially stable in blood, thanks to fast complexing with blood biomolecules (unpublished data).
Point 6. Section 3 (handling and storage) appears to be designed for the description of blood-related miRNAs. This section needs to be expanded, considering the ex-miRNAs present in other biofluids to answer the questions: are all the fluids equivalent in terms of storage conditions?, is there any general protocol for the fluid storage prior to miRNA analysis?.
Thank you for the valuable comment! We have added more information in Section 3 (handling and storage).
Point 7. Table 3 needs to be re-designed. In this current form it is very difficult to read and more important, it is almost impossible to realize what the authors want to show. I understand that the subjacent idea is to compile the comparative studies of miRNA extraction to derive some functional information for the reader. In order to achieve this goal, the table could be converted to a graphical scheme where the different methods are represented by symbols with different sizes according to their applicability in different scenarios. Remember that a typical reader of this kind of paper would always look for comparative analysis represented in a very clear and appealing form, easy to read, and to understand. The information should include all the relevant details, avoiding general descriptions of methods such as “total RNA extraction”.
Just so, this table is aimed to summarize the data of comparative studies of different miRNA isolation methods. For sure, the graphical scheme is usually more simple to perceive. However, the studies, presented in Table 3 are rather heterogeneous, they compare different methods of miRNA isolation from different biofluids, different miRNAs are assessed by different quantification method. Moreover, the date presented in the publications could not be presented in the units common for all studies. Such heterogeneity does not allow to present the data graphically.
Point 8. Section 4, should be divided into sub-sections, namely those describing the methods, technical details and applications. In this current text version, the information is very disperse and redundant sometimes.
Thank you for the valuable suggestion, the section 4 was divided into two sections (4.1 General parameters defining isolation method and 4.2 Methods of miRNA isolation from biofliuds).
Point 9. The authors would need to derive clear conclusions from their study, showing a “golden standard protocol” for extraction of miRNAs from biofluids. It would be advisable to comment also the intrinsic characteristics of some biofluids like saliva, which high glycoprotein content could interfere with the extraction procedure. The manuscript should comment these differences, proposing technical attachments to the general protocol that will be applied to specific biofluids. For instance, it would be advisable to previously remove sugars from the saliva before RNA extraction.
We thank the reviewer for the suggestion. We present the general characteristics of such biofluids in respect to their interaction with miRNAs like blood and urine which are most frequently used for miRNAs isolation at P4-6. To remove sugars is not a good idea because a dies used for their binding binds RNAs as well (obviously thanks to the fact that they are also contains polysaccharide backbone). Selection of the conditions for dissociation of nucleoprotein complexes and conditions for selective binding of miRNAs with adsorbents looks like the best choice, however, this is our own opinion.
Returning back to “golden standart” the initial phenolic extraction is considered the most universal but not without drawbacks and not always most effective. However, taking into account individual characteristics of biofluids, it is not possible to choose one specific protocol as the "gold standard".
Point 10. I would also suggest to include a specific subsection related with the so called "carriers". The authors briefly refer the use of glycogen for RNA precipitation, but there are other carriers also frequently employed like yeast tRNA or the MS2 RNA (which are just cited in a specific method described in Table 3, but not discussed alongside the text). What are the advantages of these carriers?. Would the authors advice a naive scientist in the field to use them? In which circumstances?
Yes, except from mentioned, LPA (linear polyacrylamide) could be also added in this list. We have introduced the information on the compounds used as carriers for miRNA precipitation and explanation of glycogen selection in the text (P.11). We thank the reviewer for the suggestion. The all relevant information has been added to the article.
Round 2
Reviewer 1 Report
The authors have significantly improved the manuscript. All required changes have been included. I do not have any further concerns.
Author Response
We are grateful to the referee for constructive comments and happy to meet his demands.
Reviewer 3 Report
The authors have modified the manuscript according to my suggestions and introduce substantial improvements to the original text.
I am still not convinced about the argument related with the stability of small RNAs in the blood. I am sorry to be picky, but it looks that the authors are talking about exogenous RNAs rather than the endogenous ones. The papers that the authors referred in their answer to my comments are related with modified oligonucleotides that are used to silence genes. These exogenously administered RNAs are indeed very unstable in the blood and other biofluids. This is not happening with small RNAs from endogenous origin. In my hands, circulating miRNAs maintain their levels in plasma or serum for more than 72h incubated at room temperature.
What are the reasons for this phenomenon?... I agree they can be qualified as diverse. One reason is the protection of these RNAs by encapsulation in extracellular vesicles or complexation with proteins, but other is in fact their poor reactivity with endonucleases. You have this paper as an example of this fact where the authors clearly demonstrate that miRNAs are more stable than longer mRNA molecules against RNAses in vitro:
https://bmcresnotes.biomedcentral.com/articles/10.1186/s13104-015-1114-z
Of course I would agree with the authors that there are many other factors contributing to the relative stability of miRNAs in fluids, but the truth is that if they are endogenous they are STABLE. Otherwise it would be not possible to use them as wide-range biomarkers as the authors propose.
Moreover, the RNAses that degrade miRNAs and other small RNAs are tipically EXOnucleases and not endonucleases. In consequence they are independent on the RNA sequence. The authors cannot state in their answers that "it is known that RNA stability depend on sequence for different RNAses, but as far as we know there is no data on RNA stability in depend from their size"
I would recommend the authors to take these facts into consideration before publication of the article.
Author Response
We undoubtly respect your point of view, but do not see obstacles to change our own.
Despite the authors of the manuscript you mention claim "In fact, the stabilities of these RNA classes to exposure to ribonucleases are independent from each other, with microRNA being more stable than mRNA" they consider stability not to RNase A but to RNase H, Exo T and Exo T7. But, RNase H and Exo T7 hydrolyze RNA as part of RNA:DNA duplexes; although Exo T hydrolyzes ssDNA and RNA, it is bacterial RNase and cannot exist in humans. The authors demonstrate their data on the Figure 3 which clearly demonstrate similar stability of miRNAs and mRNAs to RNase A (in addition, the authors provide data on the mean, but do not show the median, and from the pictures it is clear that the same differences between the medians will not be, few points could not be as a general regularity, moreover there are no data demonstrating statistical indicators) but higher stability of all RNA to other used enzymes (these data are actually obvious so far as RNAse H, for example, is frequently used for RNA studies and provide RNA stability in heteroduplexes).
Moreover it is known that main human RNase in blood - RNase I is at least two orders of magnitude more active that bovine RNase A against polyA and dsRNA. Actually this enzyme efficiently hydrolyzes double straining RNAs (for review see: Salvatore Sorrentino The eight human ‘‘canonical” ribonucleases: Molecular diversity, catalytic properties, and special biological actions of the enzyme proteins FEBS Letters 584 (2010) 2194–2200). RNases 2 and 3 are less presented in blood and less active against double stranded RNA and polyA and thus less important enzymes in blood.
Moreover, we have an experience in study of miRNAs stability against RNases as well as miRNAs stability and complexing in blood (but do not publish these data yet) where we do not observe higher stability of miRNA but fast complexing of synthetic ribooligonucleotides (21mers) in blood. Due to previously mentioned facts and our own experience we prefer not introduce the information regarding better stability of miRNAs to nuclease activity in circulation not to express a controversial point of view.